# Quality Assurance in Modern Gynecological HDR-Brachytherapy (Interventional Radiotherapy): Clinical Considerations and Comments

**DOI:** 10.3390/cancers13040912

**Published:** 2021-02-22

**Authors:** Tamer Soror, Frank-André Siebert, Valentina Lancellotta, Elisa Placidi, Bruno Fionda, Luca Tagliaferri, György Kovács

**Affiliations:** 1Radiation Oncology Department, University of Lübeck/UKSH-CL, 23538 Lübeck, Germany; 2Radiation Oncology Department, National Cancer Institute (NCI), Cairo University, Cairo 11796, Egypt; 3Clinic of Radiotherapy, University Hospital of Schleswig-Holstein, 24105 Campus Kiel, Germany; Frank-Andre.Siebert@uksh.de; 4UOC Radioterapia Oncologica, Dipartimento di Diagnostica per Immagini, Radioterapia Oncologica ed Ematologia, Fondazione Policlinico Universitario A. Gemelli, IRCCS, 00168 Roma, Italy; valentina.lancellotta@policlinicogemelli.it (V.L.); elisa.placidi@policlinicogemelli.it (E.P.); bruno.fionda@yahoo.it (B.F.); luca.tagliaferri@policlinicogemelli.it (L.T.); 5Università Cattolica del Sacro Cuore, Radioterapia Oncologica, Gemelli-INTERACTS, 00168 Roma, Italy; kovacsluebeck@gmail.com

**Keywords:** quality assurance, brachytherapy, interventional radiotherapy, gynecological cancers

## Abstract

**Simple Summary:**

This is a focused review discussing quality assurance during interventional brachytherapy in gynecological cancers. This topic is very large and is usually addressed from the technical and physical sides, therefore, we decided to select “hot-spots” under this large title and discuss them from the point of view of clinicians. We hope that this concise and focused review will help clinicians in improving their quality assurance protocols and draw attention to the discussed issues.

**Abstract:**

The use of brachytherapy (interventional radiotherapy) in the treatment of gynecological cancers is a crucial element in both definitive and adjuvant settings. The recent developments in high-dose rate remote afterloaders, modern applicators, treatment-planning software, image guidance, and dose monitoring systems have led to improvement in the local control rates and in some cases improved the survival rates. The development of these highly advanced and complicated treatment modalities has been accompanied by challenges, which have made the existence of quality assurance protocols a must to ensure the integrity of the treatment process. Quality assurance aims at standardizing the technical and clinical procedures involved in the treatment of patients, which could eventually decrease the source of uncertainties whether technical (source/equipment related) or clinical. This commentary review sheds light (from a clinical point of view) on some potential sources of uncertainties associated with the use of modern brachytherapy in the treatment of gynecological cancers.

## 1. Modern Gynecological Brachytherapy (Interventional Radiotherapy)

Image-guided adaptive brachytherapy (IGABT) is a crucial component in the treatment of different gynecological tumors. Despite the major recent technical advances in external beam radiation treatment (EBRT), IGABT is still superior to intensity modulated radiation treatment (IMRT) using photons or protons in terms of target coverage as well as sparing of the organ at risk (OAR) [1]. Dose escalation with IGABT significantly improves the survival of patients with cancers of the cervix in definitive settings [2,3,4,5,6] or in adjuvant settings in high risk patients [7].

Quality assurance (QA) for IGABT aims at standardizing the technical and clinical procedures involved in the treatment of patients, which could eventually decrease the source of uncertainties, both technical, such as source calibration, and clinical, such as delineation. However, dose planning and delivery are always related with the technical and physical properties of brachytherapy afterloaders [8,9,10,11,12,13], and while most of the uncertainties are related to the anatomical site of the brachytherapy as well as to the geometry of the implant, very few uncertainties are independent of the clinical aspects of implant [13].

Furthermore, QA also guarantees the reliability of the evaluation process of the treatment results, in terms of local control and treatment-related toxicity. Treatment results are usually correlated to dose parameters for the target and the OAR as obtained from the dose volume histogram (DVH), which is an important indicator for setting the target prescription dose as well as the dose limits for the OAR.

## 2. Selection of Applicator and Approach

A crucial element in the QA of IGABT is the proper selection of applicators; all clinical and technical factors should be considered before a decision is made on the use of a certain applicator for each individual patient. Appropriate applicator selection is fundamental for tumor dose coverage, which is directly linked to rates of local disease control [14]. A proper applicator should provide maximum conformity to tumor shape and volume, and, at the same time, it should be standardized and should provide efficient and consistent physical properties for successive implants [14,15].

Inappropriate applicator selection or inappropriate placement results in a poor tumor coverage which could not be compensated for by optimization, resulting in lower local disease control and lower disease-free survival. Furthermore, attempts to obtain better coverage through optimization may even result in dangerous hot spots and end in increasing toxicities [16,17].

Brachytherapy for gynecological tumors could be implemented through an intracavitary, interstitial, or hybrid (combination) procedure. In the case of vaginal-vault adjuvant brachytherapy or non-operable uterine cancer, an intracavitary application is usually selected, as is often the case also for small primary vaginal tumors. However, in large, deeply infiltrating tumors, or when involving the vulva or perineal tissue, a combined intracavitary and interstitial approach might be the most appropriate [18].

The selection of a specific approach for IGABT in cervix tumors is highly dependent on the anatomy and extent of original as well as residual disease after EBRT (± chemotherapy) [19]. The two most common commercially available applicator designs for intracavitary brachytherapy are tandem and ovoid, or tandem and ring. There is some difference between the two designs in usage and in the resulting dose distribution, however both have similar clinical results and the selection between them is user dependent [14]. In certain clinical scenarios, such as where the disease is bulky or locally advanced, or where the patients has a very narrow vaginal apex or an obliterated endocervical canal, intracavitary brachytherapy would not be appropriate. An interstitial or a hybrid applicator would be the proper selection [20]. Two commonly used interstitial templates, the Syed-Neblett template and the Martinez Universal Perineal Interstitial Template, or a vaginal cylinder and/or intrauterine tandem, could be also mounted on the template [15]. However, their proper use is mostly operator-dependent and requires specialist experience. Additionally, applicator reconstruction is technically challenging [21]. Hybrid applicators allow the insertion of interstitial needles in combination with a tandem and ovoid or a tandem and ring applicator, their use is less operator-dependent and more suitable for day procedures [15]. 

Gynecologic IGABT implants require multidisciplinary and large-volume centers. As with to all surgical procedures, user-dependent skill and experience are key factors for their success [22,23].

## 3. Image Guidance, Target Definition and Treatment Adaptation

### 3.1. Image Modality

MRI is superior in demarcating the extent of gynecological tumors and also in the delineation of the OAR, and consequently with its use better DVH value results are to be anticipated. However, although this seems an obvious advantage, studies have shown that the delineation of the OAR and doses did not differ significantly when comparing MRI and CT-based planning [24,25]. CT machines are widely available in most radiation facilities; while MRI machines are less available. MRI-based planning requires more time, logistical planning, and financial costs for IGABT and necessitates the use of MRI-compatible applicators [25]. 

Past studies have shown that high-risk clinical target volumes (HRCTVs) in cervix tumors delineated by CT (HRCTVct) are statistically different from MRI-delineated HRCTVs (HRCTVmri), even with pre-planning MRI [24,26,27,28,29]. Such volumetric differences have translated into change in HRCTV D90, while the dose to OAR shows no statistical difference. Viswanathan et al. proved that the HRCTVct D90 in CT-based plans is statistically lower than the HRCTVmri D90 in MRI guided plans, with mean doses per fraction of 6.7 Gy vs. 8.7 Gy, respectively [25]. Bhavana et al. showed similar findings, with the mean difference in dose per fraction less than 1 Gy [26].

Due to the inferiority of CT in defining the limits of tumors, especially when preceded by EBRT (± chemotherapy) as in the treatment of cancers of the cervix, using CT usually leads to an overestimation of the tumor volume especially towards the rest of uterus cranially as well as along the parametrial extensions [25,27,30,31]. The relatively larger target volumes may lead to a decrease of the dose–coverage indicators of D100 and D90 [25]. However, the impact of using CT or MRI on the size of target volumes was minimal in small tumors without parametrial extensions [27]. Moreover, a distinctive advantage of using MRI is the accurate contouring of the bowel loops and their differentiation from the uterus [25].

The use of ultrasound (US) (abdominal and or trans-rectal) in most centers is usually limited to guiding the placement of the applicator and possible interstitial needles, however combining three-dimensional abdominal or trans-rectal US images with CT has resulted in the better definition of tumor tissues and reduced overestimations of target volumes, creating treatment plans that were closer to MRI-based plans [32,33]. Using trans-rectal images solely for planning in cancers of the cervix was comparable to MRI-based planning in a small study [34]. The challenges of accurate applicator reconstruction through the use of US images without using additional tracking tools ensures the clear recommendation of QA that this approach is not practical and currently difficult [35].

### 3.2. Image-Registration

Image registration is routinely used in the modern EBRT to evaluate setup errors and to assess the need to adapt the treatment along the course of treatment. In gynecological IGABT, image registration is used for applicator reconstruction, definition, or propagation of target volume, or for dose accumulation [36]. Most radiation treatment centers use rigid image registration (RIR), however, ignoring the deformable nature of soft tissues may create unexpected sources of error and uncertainty [36]. Deformable image registration (DIR) was recently introduced through a number of innovative software programs [37,38,39]. During IGABT, an applicator-based RIR is recommended between two image modalities with the applicator in place to propagate target volumes, for example, in the case of an MRI-based first fraction to subsequent CT-based fractions or from trans-rectal US (TRUS) to CT [32,40]. DIR is generally used when registering one image series without an applicator as the diagnostic MRI images, or when registering the EBRT simulation or cone-beam CT images to another image series with an applicator (CT or MRI) for the purpose of target volume definition or dose accumulation (between IGABT fractions or between EBRT and IGABT) [41,42,43,44,45]. Extra caution should be taken when using DIR as it may result in implausible registration [36].

### 3.3. Applicator Reconstruction

Accurate applicator reconstruction is a crucial factor in decreasing the uncertainties during IGABT, as for each 1 mm that an applicator is displaced during the reconstruction there is a 5–6% change in the mean DVH parameters [46]. It is advisable that applicator reconstruction is performed on the same image study used by the radiation oncologist to delineate the target volumes and the OAR in order to avoid unnecessary errors and uncertainties resulting from the registration of different image series [19,47]. Brachytherapy applicators can be manually reconstructed or reconstructed through library applicators. Both methods were compared and revealed a limited impact on the calculated dose, however, using the library method was associated with smaller standard deviations suggesting the lesser probability of errors and uncertainties [48]. 

Although MRI-based IGABT is superior in delineating target volumes and the OAR, it may be challenging for direct applicator reconstruction as the source path (applicator lumen) is difficult to identify especially on T2 images which are recommended for delineation [19,47]. It may be necessary to use additional markers when attempting to identify the source path, thus enabling the direct reconstruction of the applicator on MRI images [49,50]. Moreover, the use of these markers is not feasible either with titanium applicators due to artifacts or in cases where the applicator lumen is too narrow to accommodate the marker [21].

In hybrid implants, where interstitial needles are combined with rigid applicators, the accurate reconstruction of the tip of each needle on MRI may be highly challenging [21]. Using the insertion length of each needle as reported by the radiation oncologist at the time of insertion may help in confirming the accurate position of each needle tip [21].

### 3.4. Contouring

In order to reduce uncertainties of contouring and maintain quality during IGABT, several factors are to be considered. High quality planning on thin-slice images enables the accurate delineation of the target as well as the OAR. If MRI is not used for planning, CT image registration with MRI and/or PET-CT improves the inter-observer contouring agreement [51,52]. GEC ESTRO recommends the use of both low (0.1–0.5 T) and high (1.0–1.5 T) field MRI machines. For cancers of the cervix, a minimum of two MRI examinations is recommended, before EBRT and at the time of IGABT [53]. The use of delineation guidelines and a dedicated institutional protocol reduces possible uncertainties and improves the agreement among radiation oncologists [54,55]. 

The GEC ESTRO has over the past years published recommendations on the delineation of target volumes in gynecological BT [18,56,57]. The ICRU report 89 emphasizes state-of-the-art methods for delineating target volumes, OAR, dose, and volume parameters for prescribing, recording, and reporting cervical cancer brachytherapy [19]. Training radiation oncologists on a selected delineation protocol with analyses of results led to improved concordance and reduced contouring uncertainties among inter- and intra-observers in IGABT [58,59,60,61].

## 4. Treatment Planning and Dose-Painting

The use of both 3-D imaging and stepping sources enables dose prescription to be directed at a volume instead of fixed geometric points, and also enables a high level of plan optimization and dose-painting [14]. In cancers of the cervix, 3-D treatment planning improved the local control rates and decreased treatment-related toxicity in comparison to 2-D planning [2,51].

Plan optimization could start off automated and be followed by manual optimization, or could be performed purely manually. For cervical treatment, for example, usually, a standard point-A-based dose distribution (pear shaped) is performed, followed by progressive manual optimization through activation or inactivation of dwell positions and varying dwell times whilst monitoring the impacts of the treatment on spatial dose distribution as well as DVH parameters for both target volumes and the OAR [14].

The GEC ESTRO recommended DVH parameters for target volumes and the OAR are widely accepted and endorsed. An important indicator for the covering of targets is the reporting of D100 and D90, which are the minimum doses covering 100% and 90% of the target volume, respectively [57]. However, the ICRU report 89 proposed D98 as an important additional parameter. Because of the unique physical properties of brachytherapy, a steep dose gradient may result in dose distribution where D90 seems acceptable, although the remaining 10% of the target volume receives a significantly lower dose [19].

High-dose volumes are of high value as they may contribute to the resulting local control rates. Heterogenicity in the dose distribution as well as steep absorbed-dose gradients are typical qualities of brachytherapy, therefore a considerable volume of the target received doses as high as 200% of the prescribed dose. After considering this unique feature of brachytherapy as well as the fraction size and the absorbed dose-rate, the calculation of the biologically effective dose across the target becomes also heterogenous reaching its maximum in the target sub-volumes located near the applicator and/or the interstitial needles. In order to evaluate these high-dose volumes, the ICRU report 89 recommended reporting of D50 [19].

## 5. Role of In-Vivo Dosimetry

In-vivo dosimetry (IVD) in gynecological brachytherapy has been typically used for dose verification [62], however, some novel methods and algorithms used IVD in identifying source localization and checking applicator reconstruction [63,64,65,66]. The difference between predicted and measured doses by IVD in hybrid gynecological brachytherapy implants was recently investigated. Differences larger than 10% between predicted and measured dose were considered unacceptable and re-imaging is recommended to detect the possible cause of error in order to avoid it in future applications [67]. Real-time treatment monitoring is a recent clinical application of the IVD which could give a clear and quick indication of errors and uncertainties during treatment delivery through the on-line submission of data during treatment progression [68].

## 6. Quality Assurance Protocol

A QA protocol regulates all the administrative procedures, technical aspects, treatment steps, and audit measures to ensure a consistent and safe fulfilment of the treatment prescription. A qualified medical physicist should be responsible for the QA protocol regarding the appropriate description, calibration, and the current source strength [69]. However, due to the complexity of brachytherapy, a more comprehensive QA protocol is needed. This protocol must include the treatment-related devices (imaging and planning, applicators, radioactive source, and afterloaders) as well as the clinical process [70].

The European Society of Therapeutic Radiology and Oncology (ESTRO) has published a booklet on quality control guidelines for brachytherapy [71]. Likewise, similar guidelines were published by the American Association of Physicists in Medicine (AAPM) [72,73], American College of Radiology (ACR) [74], and the International Atomic Energy Agency (IAEA) [75,76]. Large clinical studies published their QA protocols and dummy-runs [77,78].

Considering the available international literature and national regulations, as well as institutional resources and needs, medical physicists, radiation oncologists, radiation technologists, and all members of a brachytherapy team should work together to build an institutional QA protocol to standardize and document the clinical workflow.

## 7. Conclusions

Gynecological brachytherapy has been significantly transformed from the era of 2-D to the era of 3-D and IGABT, with this huge development being led by major technological advances over the past years. Modern IGABT is a valuable advanced and complex tool in the treatment of gynecological tumors; its proper use requires dedicated training and a solid QA protocol must cover not only the physical and technical aspects of the treatment but also the many clinical considerations in order to decrease all possible sources of uncertainties and thus improve the treatment outcome and reduce toxicity.

## Data Availability

Not applicable.

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
