# Peer review of "Quality Assurance in Modern Gynecological HDR-Brachytherapy (Interventional Radiotherapy): Clinical Considerations and Comments"

_cancers, 2021, doi:10.3390/cancers13040912_

Round 1

Reviewer 1 Report

This manuscript described Quality assurance in brachytherapy for gynecological cancer. Brachytherapy has important role in definitive radiotherapy for gynecologic cancer. The article is well written and this is interesting topics in this field. Although there are  minor concerns, I guess this work is worthy of being accepted after minor revision.

1. Authors described the recommendation of DVH parameters for target volumes, such as D100, D98 and D90. On the other hand, ICRU report 89 proposed that high-dose volumes for intracavitary brachytherapy are regarded as important because they probably contribute to the excellent local control observed, even for large-volume disease, even though there are substantial radiobiological uncertainties in calculating higher doses. Authors should discuss this.

2. Currently, Brachytherapy with various dose rates such as LDR, PDR, and HDR is being performed worldwide. However, this review is considered to be mainly related to HDR. It is recommended to add the description of the difference by dose rate, or add ‘HDR-brachytherapy’ in the title.

Author Response

  1.  

Authors described the recommendation of DVH parameters for target volumes, such as D100, D98 and D90. On the other hand, ICRU report 89 proposed that high-dose volumes for intracavitary brachytherapy are regarded as important because they probably contribute to the excellent local control observed, even for large-volume disease, even though there are substantial radiobiological uncertainties in calculating higher doses. Authors should discuss this.

Reply:

Thanks for this valuable remark. A new paragraph discussing this issue was added, lines 206 – 214.

  1.  

Currently, Brachytherapy with various dose rates such as LDR, PDR, and HDR is being performed worldwide. However, this review is considered to be mainly related to HDR. It is recommended to add the description of the difference by dose rate, or add ‘HDR-brachytherapy’ in the title.

Reply:

The title was changed, HDR-brachytherapy was added.

Reviewer 2 Report

Reviewer's report

Manuscript ID: cancers-1111112

Title:Quality assurance in modern gynecological brachytherapy (interventional radiotherapy); clinical considerations and comments

Date:2021/2/10

Reviewer's report:This is an interesting manuscript as it is a comphrehensive analysis in modern interventional gynecological brachytherapy. The use of brachytherapy (interventional radiotherapy) in the treatment of gynecological cancers is a crucial element whether in the definitive or in the adjuvant setting. The recent development of Image-guided adaptive brachytherapy (IGABT) is a crucial step in providing a more confromal and an optimal dose distribution in the treatment of gynecological cancer. Thus, obtaining a higher control rate as well as overall survival rate.This is one of the very few review articles with a thorough discussion on the new interventional technology of gynecological bracytherapy. Therefore, I'm sure this manuscript will add to a growing body of literature on the benefit of these new innovation in the treatment of gynecological cancers .

Author Response

This is an interesting manuscript as it is a comphrehensive analysis in modern interventional gynecological brachytherapy. The use of brachytherapy (interventional radiotherapy) in the treatment of gynecological cancers is a crucial element whether in the definitive or in the adjuvant setting. The recent development of Image-guided adaptive brachytherapy (IGABT) is a crucial step in providing a more confromal and an optimal dose distribution in the treatment of gynecological cancer. Thus, obtaining a higher control rate as well as overall survival rate.This is one of the very few review articles with a thorough discussion on the new interventional technology of gynecological bracytherapy. Therefore, I'm sure this manuscript will add to a growing body of literature on the benefit of these new innovation in the treatment of gynecological cancers.

Reply:

Thank you very much for your encouraging and thoughtful comment.

Reviewer 3 Report

The experts from multiple institutions present a mini-review on QA in gynecological brachytherapy. It is timely concise mini-review that highlights the important issues.

I do not think that invitation should be mentioned in the abstract.

One issue of QA was totally missed in the review: audit both internal and external, so authors should add a subchapter on this issue. Any internationally available QA programmes?

Contribution of each author should be more specifically mentioned.

typo in 3.4. paragraph title

Author Response

The experts from multiple institutions present a mini-review on QA in gynecological brachytherapy. It is timely concise mini-review that highlights the important issues.

1.

I do not think that invitation should be mentioned in the abstract.

Reply:

The invitation was removed from the simple summary.

2.

One issue of QA was totally missed in the review: audit both internal and external, so authors should add a subchapter on this issue. Any internationally available QA programmes?

Reply:

Thank you for your proposal. A dedicated chapter was added to discuss the QA-protocols and guidelines, lines 226-240.

3.

Contribution of each author should be more specifically mentioned.

Reply:

Conception and manuscript design:                  

György Kovács, Tamer Soror

Literature review:

György Kovács, Tamer Soror, Frank-André Siebert

Review of medical physics aspects:

Frank-André Siebert

Manuscript writing:

Tamer Soror, all authors

Final approval of manuscript:

All authors

Supervision and work plan

György Kovács

4.

typo in 3.4. paragraph title

Reply:

Corrected.